# Effectiveness of Positive Deviance, an Asset-Based Behavior Change Approach, to Improve Knowledge, Attitudes, and Practices Regarding Dengue in Low-Income Communities (Slums) of Islamabad, Pakistan: A Mixed-Method Study

**DOI:** 10.3390/insects13010071

**Published:** 2022-01-08

**Authors:** Muhammad Shafique, Muhammad Mukhtar, Chitlada Areesantichai, Usaneya Perngparn

**Affiliations:** 1College of Public Health Sciences, Chulalongkorn University, Bangkok 10330, Thailand; muhammad.shafique2002@gmail.com (M.S.); chitlada.a@chula.ac.th (C.A.); 2Directorate of Malaria Control, Islamabad 44000, Pakistan; mukhtarnih@gmail.com; 3Health and Social Science and Addiction Research Unit (HSSRU), Bangkok 10330, Thailand

**Keywords:** dengue, prevention, positive deviance, behavior change, community participation, slums, Pakistan

## Abstract

**Simple Summary:**

Dengue is a mosquito-borne infection caused by the *Aedes* mosquito, expanding at an alarming pace around the world. Recently, Pakistan has witnessed some major dengue outbreaks, affecting thousands of individuals across the country. As there is no specific cure or vaccine, prevention and vector control remain the key methods to avoid dengue infection. In Pakistan, dengue control activities are mainly focused on information-sharing through mass media and communication materials such as pamphlets and posters. The main challenge is a lack of community participation that can create an enabling environment for communities to follow the desired behaviors. There is a strong need to design and implement community-led behavior change approaches to ensure community participation and translate the knowledge into practices. This study was conducted to better understand the effectiveness of a community engagement approach, ‘positive deviance’, on dengue prevention and control. The study was carried out in two slums affected by the recent dengue outbreak in Islamabad, Pakistan. A total of 112 persons participated in the study, which was conducted from June–October 2020. The community discovered already-existing positive behaviors surrounding dengue prevention and control, which were shared with other participants through interactive activities. The study demonstrated positive changes in knowledge, attitudes, and practices, and could be a potential tool for dengue prevention and control programs worldwide.

**Abstract:**

Dengue is a mosquito-borne, viral disease that has emerged as a global health concern in recent years. In the absence of specific antiviral treatment and vaccines, prevention remains the key strategy for dengue control. Therefore, innovative and community-driven approaches are required to improve the vector control practices. This study applied and evaluated the positive deviance (PD) approach on dengue prevention and control in selected slums of Islamabad during June–October 2020. The two most dengue-affected slums, the Faisal colony and France colony, were purposively selected as intervention and control groups, respectively. A total of 112 participants (56 for the intervention and 56 for the control group) participated in the study. The intervention group was exposed for two months to locally identified role model behaviors through weekly interactive sessions, dengue sketch competitions, and role plays. Another two months enabled the community to practice these behaviors without any external support in order to explore the intervention’s sustainability. Three surveys were conducted: before the intervention, after two months, and after four months, to assess any changes in the knowledge, attitudes, and practices of participating communities. Results found that the PD intervention had a significant positive impact on dengue knowledge, attitudes, and practices in the intervention group. PD could offer an empowering and efficient community engagement tool for future dengue prevention and control, both in Pakistan and more globally.

## 1. Introduction

Dengue, a mosquito-borne, viral disease, has emerged as a global health concern in recent years. Dengue is prevalent in 128 countries, mostly in the tropical and subtropical regions of the world [1,2]. Dengue has four serotypes (DENV) and is primarily vectored by *Aedes aegypti* mosquitoes [3]. Dengue cases have been significantly increasing globally, with an estimated 390 million dengue infections per year [4]. Population growth, urbanization, climate change, and international travel have contributed to the rapid increase in dengue worldwide [5,6,7,8].

Dengue has spread dramatically in Pakistan in recent years [9,10]. The first confirmed dengue outbreak was reported in the economic hub of Pakistan, Karachi, in 1994 [11]. Pakistan suffered major dengue outbreaks during 2006, 2007, 2008, 2010, and 2011, which severely affected thousands of individuals and claimed hundreds of lives [12]. An estimated 24,938 dengue virus infections were recorded from 15 districts of Khyber Pakhtunkhwa in 2017 [13]. However, the worst dengue outbreak was recorded in 2019, which caused 56,000 dengue cases and claimed 95 deaths in Pakistan [14]. An estimated 43% of dengue cases were reported from Islamabad and Rawalpindi.

Currently, there is no antiviral treatment available for dengue. Although several dengue vaccines are in the clinical development process, however, it will take them years to be rolled out in Pakistan to provide protection against dengue [15,16]. Therefore, in the absence of specific treatments and vaccines, prevention remains the key method for dengue control. Therefore, local, context-appropriate, community-driven, and sustainable behavior change communication strategies are required for the effective prevention and control of dengue [17,18]. However, the key challenge to carrying out effective community-based dengue programs is the lack of community participation in vector control interventions [19]. The importance of community involvement in health and development programs has been stressed in the Alma Ata Declaration held in 1978, and thereafter has been a core goal of health planners and practitioners [20].

As the breeding of dengue vectors depends on human behavior, there is a dire need to involve local communities as active partners in the design, implementation, and monitoring activities of dengue control programs [21,22]. The active involvement of communities in the design, planning, and most important decision making creates strong ownership and acceptance which fosters the sustainability of the interventions [21,22,23,24]. This study applies positive deviance, a community engagement approach to improve vector control behaviors in Islamabad, Pakistan.

### Positive Deviance

Positive deviance (PD) is a community engagement approach to behavior change. PD was initially envisioned on nutrition studies and operationalized to improve nutrition outcomes in Vietnam. The successful nutrition experience was later replicated in over 40 countries all over the world [25,26,27,28]. The PD approach has recently been employed on a variety of public health issues which include maternal and newborn health, diabetes care, and malaria prevention and control [29,30,31].

The PD premise is that in every community, there are a few ‘positive deviant’ persons who deviate from social norms and practice uncommon behaviors that help them and their families to enjoy better health outcomes than their peers and neighbors with whom they share similar risks and resources. PD emphasizes that solutions to most health and social problems already exist in the same communities. In contrast to the need-based problem-solving approaches which look at what is missing and try to fix it, PD focuses on what is working and building on the existing strengths. These local, accessible, and acceptable solutions are then shared with other community members through an interactive implementation program to foster positive changes in their behaviors. These PD individuals or ‘role models’ have strong ownership and acceptance of their fellow community members, who can better relate to their messages and behaviors than those which are delivered by external bodies. Expert-driven approaches often fail, as the community members do not relate to those external messages and behaviors and discontinue them as soon as the externally designed and delivered intervention is complete [32]. On the other hand, the PD approach ensures a sense of belonging by identifying role models from within the community with similar resource bases and challenges. PD behaviors are simple, and therefore accessible, affordable, and replicable by the other community members facing similar risks [33]. Furthermore, active community participation throughout the process guarantees community acceptance and ownership which fosters sustainability, even years after the intervention has been completed [34].

Despite a widespread application of positive deviance as an empowering tool on a variety of public health issues, PD has never yet been employed on dengue prevention and control. PD could offer an empowering and efficient community engagement tool for dengue prevention and control. This study aims to assess the effectiveness of the PD approach on dengue prevention and control in the selected slums of Islamabad. As dengue is a newly emerging disease, there is a limited understanding and evidence of the role of communities on dengue control in Pakistan.

Therefore, this PD study will pave the way for further community engagement research and will provide rich insights to the concerned partners on the importance of community participation in dengue control and spread. 

## 2. Materials and Methods

### 2.1. Study Setting

Two low-income communities often referred to as slums, Faisal colony and France colony, were purposively selected for the study. The estimated population of Faisal colony was 4000 persons (450 households) and the estimated population of France colony was 4500 persons (500 households). In 2019, Pakistan experienced one of the worst dengue outbreaks, with 50,535 dengue cases and 83 deaths. An estimated 43% cases were reported from Islamabad and Rawalpindi. The two selected communities were among the most-affected areas of the recent dengue outbreak. 

Faisal colony was selected to receive the intervention, with France colony selected as the control group. The population of the selected slums were predominantly Christian with similar socioeconomic characteristics. The geographical distance between the two communities was around 4 KM.

### 2.2. Study Design and Sampling

A mixed method quasi-experiment was conducted in two purposively selected, high-risk dengue slums in Islamabad during June–October 2020. Convenience sampling was used to recruit 112 respondents; 56 were assigned to the intervention arm and 56 were assigned to control group. The study participants were selected based on the list of households. The households were contacted to ask for their willingness to participate in the study. One person was selected per identified household for the study. The estimated sample size was calculated using a power analysis with G*Power 3.1 [35]. The effect size was calculated using a previous study [36]. A power of 0.8 was based on the effect size of 0.59 to account for the difference in dengue knowledge between groups, which is the primary outcome. The power analysis contained four independent variables:
βBeta error, where power = (1-Beta error): 0.8αAlfa error rate: 0.05E Effect size: 0.59NSample size: 92

The total sample size included 92 participants, where 46 participants were divided into two groups. After calculating the drop-out rate of 20%, the total sample size was 112, which resulted in 56 participants per group. ANOVA: repeated measures, between–within interactions. Sample size calculation:

Effect size
$$f=\frac{\sigma_{\mu }}{\sigma }$$
$$\lambda =f^{2}\mu N_{\varepsilon }$$when
$$\mu =\frac{m}{1-\rho }$$
$$df_{1}=\left(k-1\right)\left(m-1\right)\varepsilon$$
$$df_{2}=\left(N-1\right)\left(m-1\right)\varepsilon$$

The questionnaire formerly used for a previous PD study in Cambodia was modified for this study [37]. Face-to-face interviews were conducted in the Urdu language. In June, a Knowledge, Attitude and Practices (KAP) survey was conducted to establish the benchmark in each slum. At the end of the two-month intervention, the KAP survey was repeated. In October, the KAP survey was repeated a third time to assess any changes in the knowledge, attitudes, and practices among study participants. The survey tool included questions about: (1) demographic and socioeconomic information which included age, gender, religion, marital status, education level, and the respondent’s monthly income; (2) knowledge about dengue transmission and symptoms; (3) health-seeking behaviors; (4) attitudes towards dengue; (5) personal protection measures and methods to avoid breeding sites; and (6) preferred channels of communication. The questionnaire was pretested with 30 participants for internal consistency and finalization of the tool.

For the qualitative component, eight focus group discussions (FGDs) were carried out with male and female community members to explore in depth their knowledge, perceptions, and practices regarding dengue. Ten in-depth interviews (IDIs) were conducted to identify and select the positive deviance role model behaviors from the communities. Topic guides were developed in the local language (Urdu) to conduct face-to-face interviews with the participants. The findings of the qualitative component formed the basis of the PD-informed intervention.

### 2.3. Training of Data Collectors

Local college students with some previous experience in the research were recruited for the study. Two days of training were organized, covering informed consent, research ethics, interviewing skills, probing and notetaking skills, topic guides (for the qualitative component), and the survey questionnaire. 

### 2.4. Data Management and Analysis

The data were entered in the Epi Data 3.1 software (Epi Data Association, Odense, Denmark), cleaned, and then exported to the Statistical Package for Social Science (IBM SPSS Statistics 25) for detailed analysis. During the statistical processing of data, standard methods of descriptive statistics were used. Variables of interest were tested for normality, mean and median were used to describe continuous data, and frequencies and percentages were used to describe categorical data. Chi squire (χ^2^) tests and Fisher’s Exact tests were performed to examine differences between the intervention and control group at the baseline. Based on a total number of correct answers, new variables were created to examine dengue knowledge (0–48), attitude (0–32), and practice (0–24). To the options for the attitude statement answers, ‘Strongly agree’, ‘Agree’, ‘Disagree’ and ‘Strongly disagree’ were assigned points of 4, 3, 2, and 1, respectively. For scoring purposes, negatively worded items were reverse coded. For the purposes of descriptive analysis, the answers to the statements related to dengue attitudes were collapsed into a 3-point scale (‘Agree’, ‘Don’t know, and ‘Disagree’). A repeated-measures mixed ANOVA, with one within-subjects factor (time) and one between-subjects factor (group), was conducted to compare the mean differences in total dengue knowledge, attitude, and practice scores between the intervention and control group over time, i.e., baseline, midline after two months, and end-line after another two months.

### 2.5. Ethical Consideration

The study protocol was approved by the National Bioethics Committee, Pakistan in April 2020 (Ref: No.4-87/NBC-451/20/ 2037). All the respondents were informed about the voluntary nature of the participation, possible risks and benefits, and the expected duration of the interview. Written informed consent was taken from each participant. The Government of Pakistan’s Standard Operating Procedures (SOPs) for COVID-19 were carefully followed during the interviews.

### 2.6. Positive Deviance Intervention

The positive deviance study was carried out in two phases; (1) One-week PD process; (2) Two-month implementation of PD intervention. The details of the PD implementation are as follows:

#### 2.6.1. Phase 1. PD Process (One Week)

The interactive one-week PD process helped mobilize and sensitize the communities for dengue prevention and control. The PD process enabled communities to understand the normative behaviors around dengue and discover uncommon positive deviant behaviors and strategies of role models that were already being practiced in the communities. The following activities were carried out in the PD process:Community sensitization meeting

A community sensitization meeting was held with key community stakeholders including religious leaders, teachers, and influential persons to introduce the PD concept through different interactive activities such as storytelling and conceptual games in the selected communities. Interested community members were identified as volunteers to provide support in the next step, i.e., the situation analysis (Appendix A, Community sensitization meeting).

Situation analysis

The situation analysis helped establish the normative behaviors around dengue prevention and control. A total of eight FGDs were carried out with male and female community members to explore their knowledge, perceptions, behaviors, and practices regarding dengue.

Positive deviance inquiry

The PD inquiry helped identify the PD role models from the FGDs and their local, accessible, and replicable strategies regarding dengue prevention and control. The PD behaviors were validated by observing the households to confirm the role models. A total of six role models—two male and four female community members—were identified. The role models and their family members had never been infected by dengue and were following the positive behaviors related to dengue prevention and control during the observation visits made by the data collectors (Table 1, examples of identified PD behaviors).

Community feedback session and action planning

After the PD behaviors were identified, a community feedback session was conducted at the end of the one-week PD process. The purpose of this activity was to review the PD findings with a larger audience, share and vet the PD behaviors and encourage community members to adopt these behaviors. The PD behaviors were interactively shared with the community using a cardboard box house representing the PD house. The PD behaviors were written on flip charts (large papers) with colorful sketches and inserted into the cardboard box house. The community members were invited one by one to take out the paper and read the behavior. After sharing the PD behaviors, a simple action plan was developed to explore ways to enable other community members to follow these simple behaviors (Appendix A, Role models share their PD behaviors during the feedback session).

#### 2.6.2. Phase 2. Positive Deviance Intervention (2 Months)

Phase 2 was the implementation of the PD-informed study for 2 months in the intervention area. The following activities were carried out in the PD implementation phase:Training of volunteers and IEC materials development

A one-day interactive training session was organized with the selected volunteers at the church of the Faisal colony. A total of 20 volunteers participated in the training. Participatory techniques such as brainstorming, group discussions, role plays, and conceptual games were used during the training sessions. Half of the training was allocated for the development of local sketches of the identified PD behaviors to be used in the local information, education, and communication (IEC) materials. After completion of the exercise, the best sketch was used in the IEC material to reinforce the key messages. 

Interactive PD sessions

PD sessions were organized by the trained volunteers in the intervention group on a weekly basis. The PD role models were also present in the PD sessions and shared their local, simple behaviors and strategies to avoid dengue, which served as social proof for the other community members. Role plays, storytelling, and locally made IEC materials were used in these sessions. On average, 20 persons participated in each session (Appendix A, PD health education session with female community members).

PD Seminar

An interactive advocacy event was organized after 6 weeks of intervention implementation. The main objective of the seminar was to acknowledge the volunteers, reinforce the key messages, and celebrate their achievements. The PD seminar was attended by most of the community members from the intervention group. The community was involved in the preparations at least one week before the seminar. The following were the main activities of the PD seminar:

Illustration competition

A sketch competition was organized to reinforce the dengue prevention and control messages. Communities were informed about the sketch competition at least one week in advance with instructions on how to prepare the dengue sketches. These colorful sketches were displayed at the seminar venue. The audience went through all the pictures and reviewed the messages on each sketch. The three best sketches were selected for the prizes (Appendix A, an illustration made by community member for the sketch competition).

Quiz competition

A quiz competition was organized during the seminar. Key questions regarding dengue were written on small pieces of paper, wrapped, and put in a basket. The seminar audience was actively engaged in the quiz competition. On giving the correct answer, the person received a small souvenir such as a bar of soap. The purpose of this segment was to reinforce the dengue messages in an interesting and engaging manner.

## 3. Results

### 3.1. Sociodemographics at the Baseline

A total of 112 participants were recruited for the study: 56 for the intervention group and 56 for the control group at baseline. At the midline (after two months) and endline (after four months), the number of participants in the control group decreased by two.

At the baseline, there were no statistically significant differences found in sociodemographic characteristics such as sex, age, marital status, religion, education, occupation, and average monthly income between intervention and control groups (Table 2).

### 3.2. Dengue Knowledge at Baseline

An equal number of respondents in the control and intervention groups knew that dengue is mosquito-transmitted (67.9%). There was a statistically significant difference in knowledge of three or more dengue symptoms between groups at the baseline where 53.6% of respondents from the intervention group knew three or more symptoms compared to 19.6% of respondents from the control group. However, no statistically significant differences were found between groups in terms of knowing about mosquito breeding sites and mosquito and dengue prevention methods. An independent sample t-test revealed that there were no statistically significant differences between the intervention and control group at the baseline in the total knowledge score, as well as in the total attitude and practice scores (Table 3).

### 3.3. Knowledge about Dengue at the Baseline, Midline, and Endline

An improvement in knowledge over time was observed for almost all examined variables, even in the control group. At the baseline, 67.9% of respondents from the intervention and control group knew that dengue is mosquito-transmitted compared to 100% of respondents from the intervention group and 83.3% of respondents from the control group at the endline. At the endline, 96.4% of respondents from the intervention and 64.8% of respondents from the control group knew that dengue mosquitoes most often bite during the day, compared to 26.8% of respondents from the intervention group and 17.9% of respondents from the control group at the baseline. Furthermore, 53.6% of respondents from the intervention and 19.6% of respondents from the control group knew three or more dengue symptoms at the baseline, compared to 96.4% of respondents from the intervention group and 70.4% of respondents from the control group at the endline (Table 4).

### 3.4. Knowledge Score Comparison between Intervention vs. Control Groups

Results showed that there was a significant main effect of time (F(1.74,187.94) = 88.492, *p* < 0.001, ηp2 = 0.450) and group (F(1,108) = 81.518, *p* < 0.001, ηp2 = 0.430), on knowledge scores. In addition, there was a significant interaction between group and time (F(1.74,187.94) = 19.037, *p* < 0.001, ηp2 = 0.150). For the pairwise comparisons, Bonferroni-adjusted paired t-tests were performed. Statistically significant differences in dengue knowledge between the control (M = 8.93, SD = 3.107) and intervention (M = 10.09, SD = 3.549) group were not found at the baseline (*p* = 0.071). After two months of the positive deviance intervention, the intervention group (M = 19.00, SD = 6.093) had a better statistically significant improvement regarding dengue knowledge compared to the control group (M = 13.13, SD = 4.953) (*p* < 0.001). Furthermore, after another two months at the end line, knowledge regarding dengue transmission, prevention practice, and symptoms not only persisted but continued to improve with statistically significant differences between the control group (M = 14.30, SD = 4.944) and intervention group (M = 25.00, SD = 9.607) (*p* < 0.001). Estimated marginal means are visualized in the profile plot, where an increase in knowledge over time can be seen in both groups, especially in the intervention group (Figure 1.).

### 3.5. Attitude towards Dengue at Baseline, Midline, and Endline

Attitude towards dengue was good overall among the respondents from the intervention and control groups. The vast majority of respondents from both groups agreed that dengue is a serious infection, that removing empty containers can protect from dengue infection, that using bed nets, repellents, and long sleeves can protect from mosquito bites, and that communities should participate in controlling dengue (Table 5).

### 3.6. Attitude Scores Comparison between Intervention and Control Groups 

Results showed that there was a significant main effect of time (F(2,216) = 25.431, *p* < 0.001, ηp2 = 0.191) and a non-significant main effect of group (F(1,108) = 0.538, *p* = 0.465, ηp2 = 0.005), on attitude scores. There was a significant interaction between group and time (F(2,216) = 4.577, *p* = 0.011, ηp2 = 0.041). For the pairwise comparisons, Bonferroni-adjusted paired t-tests were performed. Statistically significant differences in dengue attitude between the control group (M = 24.52, SD = 4.064) and intervention (M = 24.21, SD = 3.329) group were not found at the baseline (*p* = 0.668). Furthermore, at the midline, after two months and positive deviance intervention, statistically significant differences in dengue attitude between the control group (M = 26.39, SD = 2.695) and intervention (M = 25.84, SD = 2.492) group were also not found (*p* = 0.269). After another two months at the end line, attitudes towards dengue improved significantly in the intervention group (M = 28.34, SD = 3.604) compared to the control group (M = 26.52, SD = 4.343) (*p* = 0.018).

Estimated marginal means of dengue attitudes are visualized in the profile plot, where an increase in dengue attitudes between baseline and midline for the intervention and control group is quite similar, but between midline and end line, control-group attitudes remain almost horizontal compared to increased intervention-group attitudes (Figure 2).

### 3.7. Dengue Prevention and Control Practices at Baseline, Midline and Endline

At the endline, an improvement in practice was observed related to dengue preventive methods, especially in the intervention group, such as sleeping under a bed net during the day, using insecticide spray, repellent, mosquito coil, and smoke to drive away mosquitoes. At the endline, 53.6% of respondents from the intervention group were covering all water containers, compared to 10.7% at the baseline. At the endline, 55.4% of respondents from the intervention group were changing the storage water once a week compared to 25% at the baseline. Interestingly, an improvement in practice was observed not only in the intervention group but also in the control group (Table 6).

### 3.8. Practice Score Comparison between Intervention and Control Groups

Results showed that there was a significant main effect of time (F(2,216) = 45.019, *p* < 0.001, ηp2 = 0.294) and group (F(1,108) = 20.070, *p* < 0.001, ηp2 = 0.157), on practice scores. In addition, there was a significant interaction between group and time (F(2,216) = 18.117, *p* < 0.001, ηp2 = 0.144). For the pairwise comparisons, Bonferroni-adjusted paired t-tests were performed. Statistically significant differences in dengue practice between the control group (M = 10.57, SD = 1.159) and intervention (M = 10.21, SD = 1.615) group were not found at the baseline (*p* = 0.184). Furthermore, at the midline, after two months and positive deviance intervention, statistically significant differences in dengue practice between the control group (M = 11.44, SD = 1.525) and intervention (M = 12.09, SD = 2.109) group were also not found (*p* = 0.070). After another two months at the endline, practice regarding dengue improved significantly in the intervention group (M = 13.77, SD = 2.216) compared to the control group (M = 11.37, SD = 1.629) (*p* < 0.001) (Table 5).

The estimated marginal means of dengue practice are visualized in the profile plot, where an increase in dengue practice between baseline and midline is quite similar for the intervention and control group, but between midline and endline control-group practice remains almost horizontal, compared to increased intervention-group practice (Figure 3).

## 4. Discussion

To our knowledge, this is the first study that has applied and evaluated the positive deviance approach on dengue prevention and control. The study was aimed to determine the effectiveness of the PD approach on dengue prevention and control in the urban slums of Islamabad.

The study revealed that there were significant changes in the knowledge, attitudes, and practices in the intervention group compared to the control group. After two months of intervention, the intervention group demonstrated a statistically significant improvement in dengue knowledge compared to the control group. Furthermore, after another two months at the endline, knowledge regarding dengue transmission, prevention practices, and symptoms not only persisted but continued to improve, with statistically significant differences between the intervention and control group. Interestingly, there were also some improvements in knowledge in the control group, which could be attributed to the Directorate of Malaria Control Pakistan, which conducted health promotion activities in all the affected communities after the dengue outbreak. There could also be some possibilities of contamination, as some of the members of the intervention group were frequently visiting the control group area to meet their relatives.

After the first two months of intervention, there were no significant changes observed in attitudes and practices of the intervention communities. After another two months, the evaluation showed that there were significant changes in the attitudes and practices of the intervention community compared to the control community. The improvements in attitudes and practices in the intervention group can be attributed to the PD-informed behavior change intervention, which transformed knowledge into practices in the intervention group, which is validated by a behavior change intervention study conducted in Cambodia [38]. Despite the improvement in knowledge within the control group, no changes were observed in the attitudes and practices in the control group at the endline, which validated Park Lloyd’s argument that knowledge alone is not enough to bring about positive changes in practices [39]. This further strengthens the notion that creating awareness alone is not sufficient unless the enabling environment is created through community participation at the household and community level for effective prevention and control of dengue [40]. Since the Alma Ata conference, community participation has been regarded as a vital element of primary healthcare programs by the World Health Organization [41]. However, unfortunately in most cases, participation remained expert-driven or top-down, where outsiders instructed communities on how to tackle the health or vector control problems [42]. It is uncommon that the community is considered as a partner and is engaged in the design, implementation, and evaluation of the health program.

In the PD study, the community was considered as an active partner, where they led the design and planning, and had a role in decision-making, which helped create interest, acceptance, and sustained community participation (no drop-out in intervention group) throughout the PD intervention, which is consistent with previous studies [20,21,22]. Equity and equality of participation were also taken seriously, ensuring that people from all segments of the community had equal opportunity to participate in the intervention [43]. The PD intervention was developed based on formative research understanding the normative behaviors around dengue. The PD inquiry (in-depth interviews) helped identify the already existent local, accessible, and easily replicable solutions which were promoted through actual role models to the other community members during the two months of intervention. The PD behaviors and messages were shared by the identified role models via storytelling, which served as social proof for the community members as they believed, *“if he/she can do it, why can’t I”*. The Information, Education, and Communication (IEC) materials were also developed by the community members using colorful sketches on flip charts (large paper) which had strong ownership of the communities. The previous studies also verified the fact that culturally sensitive and context-specific behavior change communication approaches were very effective at improving awareness and practices on dengue prevention and control [17,38,43]. Many randomized controlled trials also demonstrated positive outcomes in reducing entomological indicators, simply because the community was seriously engaged and the interventions were tailored to the local context [23,44,45,46]. The PD intervention was flexible, culturally appropriate, and led by the community. A similar study validated that a well-informed and context-appropriate intervention guarantees positive behavior changes [47].

The PD approach has tremendous potential as an effective behavior change and community engagement tool for dengue prevention and control. The approach has been very successful in engaging communities and received very positive feedback from a small-scale evaluation of a malaria study conducted in Cambodia [30]. The PD approach was well-received in the community and produced a significant amount of interest, motivation, and empowerment among participants, which in turn improved the dengue-related outcomes in the intervention community. The real promise of the approach is the signs of sustainability of the intervention, with the dengue prevention and control behaviors sustained after two months of the end of the intervention (in the intervention group).

Another key reason for the active participation of the community in the study was the community-based interactive activities, such as small competitions. For example, the community members were engaged in competitions to develop sketches/drawings or develop songs on those messages or behaviors they heard in the last two months. This generated great interest, as they involved their families and neighbors in the competition to come up with good drawings and songs, which in turn mobilized the community and reinforced dengue behaviors at the community level. Community members were acknowledged with some small souvenirs (worth 1 USD per souvenir) for their best drawings and songs in front of the large community, which boosted their confidence and maintained their motivation throughout the intervention. Therefore, as Atkinson’s study discussed, small incentives such as souvenirs should be considered to enhance participation and motivation in the vector control programs [48].

## 5. Limitations

The study was conducted on a small scale due to scarce financial resources. As the study used quasi-experiments using convenience sampling methodology, the findings therefore cannot be generalized; however, the study still provides some excellent insights on the process and evaluation of the approach. As the study was conducted in slum areas, the findings may only be relevant to a similar context.

## 6. Conclusions

The findings revealed that the positive deviance study had a significant impact on dengue knowledge, attitudes, and practices in the target communities, even in the short span of the implementation period. The study recommends that well-informed and community-led behavior change approaches are needed to ensure community participation and sustained behavioral changes in dengue prevention and control programs. The study demonstrated that locally identified solutions and community-made IEC materials had strong ownership and acceptance from the community. PD is an ‘inside-out’ community-led behavior change intervention that ensures active community participation throughout the process, which is a key requirement of vector control programs. Therefore, PD should be further replicated and scaled up to better determine the effectiveness of the approach. Positive deviance could be a potential behavior change tool to be adopted for dengue prevention and control in Pakistan and elsewhere.

## Figures and Tables

**Figure 1 insects-13-00071-f001:**
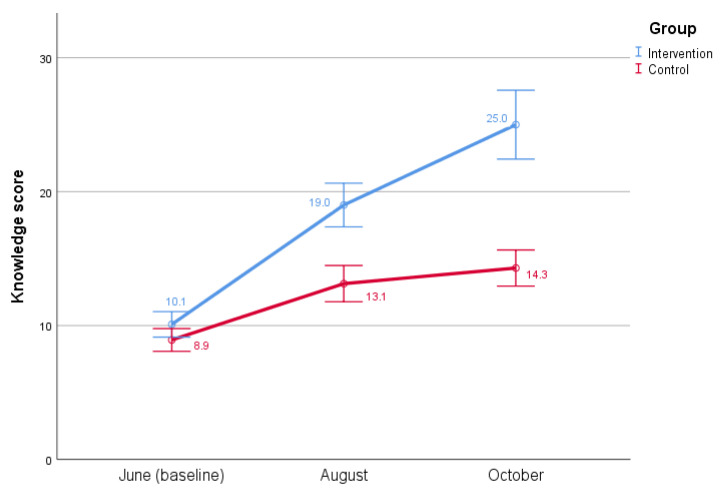
Change in mean dengue knowledge scores during three different times in 2020 in two low-income communities in Islamabad. Maximum knowledge score was 48.

**Figure 2 insects-13-00071-f002:**
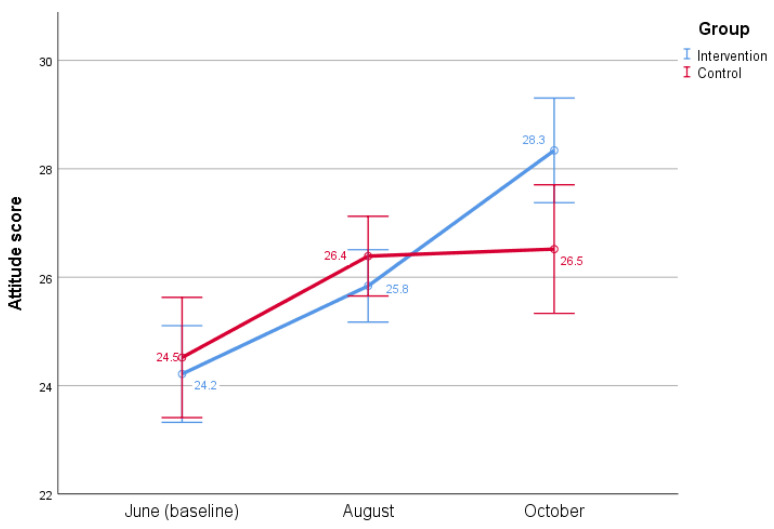
Change in mean dengue attitude scores during three different times in 2020 in two low-income communities in Islamabad. Maximum attitude score was 32.

**Figure 3 insects-13-00071-f003:**
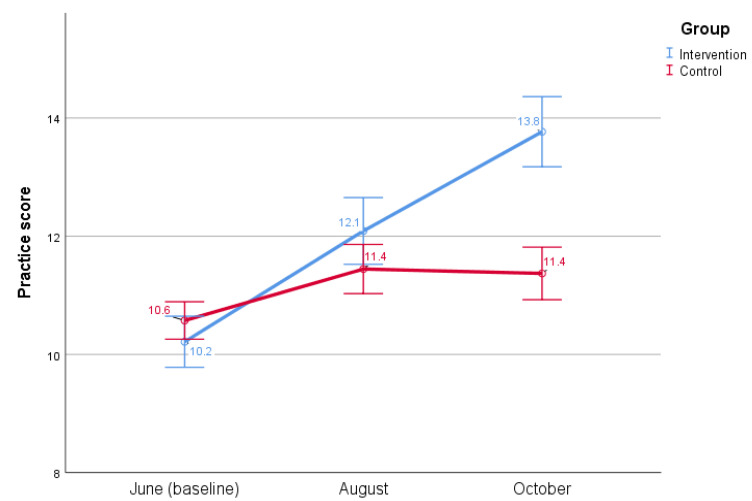
Change in mean dengue practice scores during three different times in 2020 in two low-income communities in Islamabad. Maximum practice score was 24.

**Table 1 insects-13-00071-t001:** Positive deviant behaviors identified during the PD process in Islamabad 3.

Desired Behaviors	Positive Deviance Behaviors
Knowledge	Correct knowledge of the dengue vector and mosquito biting time
	Knowledge of dengue mosquito’s breeding places inside and outside the house
Avoid water storage	A housewife does not store water. She uses it immediately so that the dengue mosquitoes do not breed in the clean water
Change the water in plants	A housewife changes the water in her plants every day to ensure no mosquitoes breed inside the plants
Cover the water containers	A housewife covers all the water tanks and water containers to avoid mosquito breeding
Clean the water tank	A male community member covers his water tank and cleans it on regular basis to avoid any mosquito breeding
	A female community member cleans her water tank with a brush and soap twice a week to avoid mosquito breeding
Bury the old bottles and tins	A housewife collects the old bottles and tins and buries them outside the house to avoid mosquitoes breeding in them
Clean the tray under refrigerator	A female community member cleans the tray which lies under the refrigerators to avoid the breeding of mosquitoes in it
Change the water in the water cooler fan	A female community member changes the water in her watercooler fan on daily basis to avoid mosquito growth inside it
Healthcare seeking	A female community member knows the signs and symptoms of dengue fever and seeks treatment as soon as she suspects dengue
Personal protection	A mother ensures that her children wear full-sleeved clothes during the day to avoid mosquito bites
	A mother keeps her children sleeping under bed net during the day to avoid mosquito bites
	A father ensures that his children wear long-sleeved clothes to avoid mosquito bites during the day

**Table 2 insects-13-00071-t002:** Sociodemographic characteristics of the intervention and control groups surveyed (baseline).

Characteristics (N = 112)	Intervention Group		Control Group		
	n (%)	Median (Range)	n (%)	Median (Range)	*p*-Value *
Sex					
Female	47 (83.9)	-	42 (75.0)	-	1.000
Male	9 (16.1)	-	14 (25.0)	-	
Age		31.0 (18–58)		30.0 (18–55)	
<30	24 (42.9)	24.0 (18–28)	25 (44.6)	21.0 (18–29)	0.789
≥30	32 (57.1)	35.5 (30–58)	31 (55.4)	37.0 (30–55)	
Marital status					
Single	15 (26.8)	-	13 (23.2)	-	1.000
Married	41 (73.2)	-	43 (76.8)	-	
Religion					
Christian	56 (100)	-	56 (100)	-	
Education					
Primary school (1–5)	5 (8.9)	-	7 (12.5)	-	0.285
Secondary school (5–9)	6 (10.7)	-	11 (19.6)	-	
High school (10)	14 (25.0)	-	19 (33.9)	-	
Intermediate-FA	5 (8.9)	-	3 (5.4)	-	
Bachelor-BA	1 (1.8)	-	1 (1.8)	-	
Masters-MA	0 (0)	-	2 (3.6)	-	
No formal education	25 (44.6)	-	13 (23.2)	-	
Occupation					
Unemployed	15 (26.8)	-	15 (26.8)	-	0.114
Government job	3 (5.4)	-	1 (1.8)	-	
Private job	11 (19.6)	-	14 (25.0)	-	
Street vendor	1 (1.8)	-	2 (3.6)	-	
Housewife	24 (42.9)	-	24 (42.9)	-	
Others	2 (3.6)	-	0 (0)	-	
Average monthly income (Rupees)					
<25,000	36 (64.3)	-	31 (55.4)	-	0.896
25,000–50,000	11 (19.6)	-	16 (28.6)	-	
50,000–75,000	4 (7.1)	-	2 (3.6)	-	
>100,000	1 (1.8)	-	0 (0)	-	
Don’t know	4 (7.1)	-	7 (12.5)	-	

* χ^2^ test/Fisher’s Exact test.

**Table 3 insects-13-00071-t003:** Dengue knowledge, attitudes, and practice based on intervention and control groups (baseline). “n” indicates the number of persons that answered ‘yes’ to the question).

	n	%	n	%	
Baseline	Intervention Group (n = 56)		Control Group (n = 56)		*p*-Value
Knowledge					
Dengue is mosquito-transmitted	38	67.9	38	67.9	1.000 *
Knows 3 or more dengue symptoms	30	53.6	11	19.6	*p* < 0.001 *
Knows 1 or more mosquito breeding sites inside the house	47	83.9	44	78.6	0.629 *
Knows 1 or more mosquito breeding sites outside the house	45	80.4	44	78.6	1.000 *
Knows 1 or more mosquito breeding prevention methods	48	85.7	50	89.3	0.776 *
Knows 1 or more dengue prevention methods	52	92.9	52	92.9	1.000 *
Total knowledge score (0–48)	Mean	SD	Mean	SD	0.062 **
	10.09	3.549	8.91	3.053	
Attitude					
Total attitude score (0–32)	Mean	SD	Mean	SD	0.627 **
	24.21	3.329	24.55	4.004	
Practice					
Total practice score (0–24)	Mean	SD	Mean	SD	0.156 **
	10.21	1.615	10.63	1.169	

* χ^2^ test; ** Independent samples *t*-test.

**Table 4 insects-13-00071-t004:** Knowledge about dengue transmission, prevention practice, and symptoms at the baseline, midline, and endline. ‘I’ means intervention group and ‘C’ means control group.

Knowledge	Baseline		Midline		Endline	
	n (%)		n (%)		n (%)	
	I Group	C Group	I Group	C Group	I Group	C Group
How is dengue transmitted?						
Mosquito	38 (67.9)	38 (67.9)	56 (100)	44 (81.5)	56 (100)	45 (83.3)
What type of mosquito causes dengue fever?						
Aedes	5 (8.9)	6 (10.7)	38 (67.9)	20 (37)	50 (89.3)	31 (57.4)
When do dengue mosquitoes most often bite?						
Bite during the day	15 (26.8)	10 (17.9)	49 (87.5)	19 (35.2)	54 (96.4)	35 (64.8)
Bite during the night time	25 (44.6)	34 (60.7)	7 (12.5)	29 (53.7)	2 (3.6)	19 (35.2)
Other	2 (3.6)	0 (0)	0 (0)	0 (0)	0 (0)	0 (0)
Don’t know	14 (25)	12 (21.4)	0 (0)	6 (11.1)	0 (0)	0 (0)
Where do *Aedes* mosquitoes usually breed inside the house?						
In the trays under the fridge	3 (5.4)	1 (1.8)	6 (10.7)	3 (5.6)	8 (14.3)	5 (9.3)
In the flower pot trays	2 (3.6)	1 (1.8)	7 (12.5)	2 (3.7)	9 (16.1)	2 (3.7)
In the water containers	35 (62.5)	22 (39.3)	30 (53.6)	35 (64.8)	31 (55.4)	18 (33.3)
In the open water tanks	8 (14.3)	22 (39.3)	30 (53.6)	16 (29.6)	34 (60.7)	34 (63)
Dirty environment	2 (3.6)	1 (1.8)	1 (1.8)	0 (0)	0 (0)	0 (0)
Don’t know	6 (10.7)	11 (19.6)	0 (0)	7 (13)	0 (0)	3 (5.6)
Knows 1 or more breeding sites inside the house	47 (83.9)	44 (78.6)	54 (96.4)	47 (87)	55 (98.2)	51 (94.4)
Where do *Aedes* mosquitoes usually breed outside the house?						
In the flower leaves	9 (16.1)	15 (26.8)	14 (25)	12 (22.2)	19 (33.9)	8 (14.8)
In the old tires	6 (10.7)	1 (1.8)	13 (23.2)	4 (7.4)	20 (35.7)	1 (1.9)
In the roof gutter	12 (21.4)	5 (8.9)	17 (30.4)	10 (18.5)	11 (19.6)	3 (5.6)
In the empty cans, shells	22 (39.3)	27 (48.2)	38 (67.9)	23 (42.6)	51 (91.1)	43 (79.6)
Dirty water	2 (3.6)	0 (0)	0 (0)	0 (0)	0 (0)	0 (0)
Don’t know	9 (16.1)	11 (19.6)	1 (1.8)	14 (25.9)	0 (0)	3 (5.6)
Knows 1 or more breeding sites outside the house	45 (80.4)	44 (78.6)	54 (96.4)	40 (74.1)	56 (100)	51 (94.4)
How can you prevent mosquitoes from breeding?						
Using insecticide in water	9 (16.1)	12 (21.4)	31 (55.4)	14 (25.9)	35 (62.5)	16 (29.6)
Changing stored water frequently	7 (12.5)	11 (19.6)	24 (42.9)	15 (27.8)	28 (50)	13 (24.1)
Turning containers upside down	10 (17.9)	21 (37.5)	31 (55.4)	20 (37)	40 (71.4)	17 (31.5)
Putting covers on water jars	27 (48.2)	20 (35.7)	45 (80.4)	28 (51.9)	34 (60.7)	22 (40.7)
Burning or burying empty cans, shells	3 (5.4)	3 (5.4)	14 (25)	7 (13)	23 (41.1)	10 (18.5)
Spraying insecticide	14 (25)	11 (19.6)	15 (26.8)	10 (18.5)	33 (58.9)	20 (37)
Clean the household	7 (12.5)	3 (5.4)	0 (0)	0 (0)	0 (0)	0 (0)
Don’t know	3 (5.4)	2 (3.6)	2 (3.6)	3 (5.6)	0 (0)	0 (0)
Knows 1 or more mosquito breeding prevention methods	48 (85.7)	50 (89.3)	55 (98.2)	51 (94.4)	56 (100)	54 (100)
How can you prevent dengue?						
Use mosquito net during the day	5 (8.9)	9 (16.1)	15 (26.8)	14 (25.9)	22 (39.3)	9 (16.7)
Wear long sleeves/long pants	23 (41.1)	22 (39.3)	48 (85.7)	37 (68.5)	52 (92.9)	42 (77.8)
Use mosquito repellent	15 (26.8)	22 (39.3)	22 (39.3)	26 (48.1)	33 (58.9)	22 (40.7)
Use insecticide spray	34 (60.7)	36 (64.3)	27 (48.2)	31 (57.4)	41 (73.2)	32 (59.3)
Cut down bushes near the house	1 (1.8)	1 (1.8)	6 (10.7)	2 (3.7)	14 (25)	6 (11.1)
Have children play far from mosquito breeding area	3 (5.4)	0 (0)	3 (5.4)	0 (0)	14 (25)	1 (1.9)
Use mosquito coils during the day	4 (7.1)	3 (5.4)	9 (16.1)	2 (3.7)	21 (37.5)	3 (5.6)
Keep household environment clean	4 (7.1)	0 (0)	4 (7.1)	4 (7.4)	26 (46.4)	12 (22.2)
Install screens on windows/doors	1 (1.8)	0 (0)	1 (1.8)	0 (0)	16 (28.6)	1 (1.9)
Keep clothes tidy	0 (0)	0 (0)	0 (0)	0 (0)	18 (32.1)	2 (3.7)
Use fan	0 (0)	0 (0)	0 (0)	0 (0)	2 (3.6)	1 (1.9)
Don’t know	4 (7.1)	2 (3.6)	1 (1.8)	2 (3.7)	0 (0)	1 (1.9)
Knows 1 or more dengue prevention methods	52 (92.9)	52 (92.9)	55 (98.2)	52 (96.3)	56 (100)	53 (98.1)
What are the symptoms of dengue?						
High fever	40 (70.1)	37 (66.1)	52 (92.9)	42 (77.8)	55 (98.2)	45 (83.3)
Headache	16 (28.6)	12 (21.4)	34 (60.7)	26 (48.1)	45 (80.4)	29 (53.7)
Chills	5 (8.9)	0 (0)	24 (42.9)	14 (25.9)	35 (62.5)	17 (31.5)
Nausea/Vomiting	12 (21.4)	4 (7.1)	31 (55.4)	17 (31.5)	36 (64.3)	19 (35.2)
Rash	13 (23.2)	3 (5.4)	24 (42.9)	9 (16.7)	22 (39.3)	7 (13)
Muscle and joint pain	11 (19.6)	11 (19.6)	26 (46.4)	16 (29.6)	37 (66.1)	15 (27.8)
Bleeding	6 (10.7)	2 (3.6)	25 (44.6)	7 (13)	35 (62.5)	12 (22.2)
Diarrhea	0 (0)	0 (0)	5 (8.9)	1 (1.9)	1 (1.8)	0 (0)
Eye pain	3 (5.4)	0 (0)	12 (21.4)	2 (3.7)	25 (44.6)	1 (1.9)
Don’t know	12 (21.4)	18 (32.1)	3 (5.4)	9 (16.7)	0 (0)	7 (13)

**Table 5 insects-13-00071-t005:** Attitude towards dengue disease at the baseline, midline, and endline.

Attitude	Agree		Don’t Know		Disagree	
	n (%)		n (%)		n (%)	
	I Group	C Group	I Group	C Group	I Group	C Group
Dengue is a serious illness?						
Baseline	56 (100)	53 (94.6)	0 (0)	0 (0)	0 (0)	3 (5.4)
Midline	55 (98.2)	54 (100)	0 (0)	0 (0)	1 (1.8)	0 (0)
Endline	56 (100)	51 (94.4)	0 (0)	1 (1.9)	0 (0)	2 (3.7)
Dengue is a transmissible disease?						
Baseline	47 (83.9)	39 (69.6)	0 (0)	2 (3.6)	9 (16.1)	15 (26.8)
Midline	50 (89.3)	46 (85.2)	0((0)	1 (1.9)	6 (10.7)	7 (13)
Endline	51 (91.1)	44 (81.5)	0 (0)	0 (0)	5 (8.9)	10 (18.5)
You are at risk of getting dengue?						
Baseline	37 (66.1)	40 (71.4)	7 (12.5)	6 (10.7)	12 (21.4)	10 (17.9)
Midline	49 (87.5)	37 (68.5)	2 (3.6)	3 (5.6)	5 (8.9)	14 (25.9)
Endline	46 (82.1)	38 (70.4)	4 (7.1)	5 (9.3)	6 (10.7)	11 (20.4)
Dengue fever can be prevented easily?						
Baseline	34 (60.7)	39 (69.6)	1 (1.8)	0 (0)	21 (37.5)	17 (30.4)
Midline	44 (78.6)	40 (74.1)	0 (0)	2 (3.7)	12 (21.4)	12 (22.2)
Endline	47 (83.9)	34 (63)	0 (0)	0 (0)	9 (16.1)	20 (37)
Can removing empty containers protect you from dengue infection?						
Baseline	49 (87.5)	48 (85.7)	0 (0)	1 (1.8)	7 (12.5)	7 (12.5)
Midline	56 (100)	54 (100)	0 (0)	0 (0)	0 (0)	0 (0)
Endline	55 (98.2)	52 (96.3)	0 (0)	0 (0)	1 (1.8)	2 (3.7)
Using bed nets, repellents, and long sleeves can protect from mosquito bites?						
Baseline	55 (98.2)	54 (96.4)	0 (0)	0 (0)	1 (1.8)	2 (3.6)
Midline	56 (100)	53 (98.1)	0 (0)	0 (0)	0 (0)	1 (1.9)
Endline	56 (100)	52 (96.3)	0 (0)	0 (0)	0 (0)	2 (3.7)

**Table 6 insects-13-00071-t006:** Dengue practice at the baseline, midline, and endline.

Practice	Baseline		Midline		Endline	
	n (%)		n (%)		n (%)	
	I Group	C Group	I Group	C Group	I Group	C Group
What do you do to prevent dengue?						
Nothing	2 (3.6)	7 (12.5)	3 (5.4)	2 (3.7)	4 (7.1)	0 (0)
Sleep under bed net during the day	9 (16.1)	16 (28.6)	15 (26.8)	22 (40.7)	29 (51.8)	18 (33.3)
Use fan to prevent mosquito bites	9 (16.1)	17 (30.4)	10 (17.9)	22 (40.7)	17 (30.4)	10 (18.5)
Use insecticide spray	40 (71.4)	41 (73.2)	47 (83.9)	41 (75.9)	53 (94.6)	49 (90.7)
Use repellent	30 (53.6)	26 (46.4)	47 (83.9)	38 (70.4)	51 (91.1)	32 (59.3)
Use mosquito coil	25 (44.6)	22 (39.3)	42 (75)	28 (51.9)	51 (91.1)	30 (55.6)
Use smoke to drive away mosquitoes	5 (8.9)	1 (1.8)	8 (14.3)	5 (9.3)	13 (23.2)	2 (3.7)
Cover all water containers	6 (10.7)	3 (5.4)	21 (37.5)	6 (11.1)	30 (53.6)	15 (27.8)
Change water in trays under the fridge	0 (0)	0 (0)	7 (12.5)	0 (0)	11 (19.6)	2 (3.7)
Destroy or burn unused containers	1 (1.8)	0 (0)	3 (5.4)	0 (0)	18 (32.1)	2 (3.7)
Do you keep covers on the water containers in the home?						
Yes	55 (98.2)	54 (96.4)	56 (100)	53 (98.1)	56 (100)	54 (100)
Please can I observe some of the containers?						
Covers observed on all containers	45 (80.4)	44 (78.6)	51 (91.1)	48 (88.9)	53 (94.6)	53 (98.1)
Covers observed on some containers	10 (17.9)	9 (16.1)	5 (8.9)	5 (9.3)	3 (5.4)	1 (1.9)
No covers observed	1 (1.8)	3 (5.4)	0 (0)	1 (1.9)	0 (0)	0 (0)
How often do you change the storage water?						
Once a week	14 (25)	18 (32.1)	21 (37.5)	20 (37)	31 (55.4)	30 (55.6)
More than once a week	34 (60.7)	37 (66.1)	31 (55.4)	34 (63)	24 (42.9)	22 (40.7)
Twice per month	2 (3.6)	0 (0)	2 (3.6)	0 (0)	0 (0)	0 (0)
Once a month	0 (0)	1 (1.8)	2 (3.6)	0 (0)	1 (1.8)	2 (3.7)
Never	4 (7.1)	0 (0)	0 (0)	0 (0)	0 (0)	0 (0)
Don’t know	2 (3.6)	0 (0)	0 (0)	0 (0)	0 (0)	0 (0)
How often do you clean the water containers?						
Every day	43 (76.8)	44 (78.6)	45 (80.4)	43 (79.6)	35 (62.5)	36 (66.7)
Once a week	8 (14.3)	9 (16.1)	10 (17.9)	11 (20.4)	20 (35.7)	10 (18.5)
Once a month	4 (7.1)	2 (3.6)	0 (0)	0 (0)	0 (0)	8 (14.8)
Occasionally	1 (1.8)	0 (0)	0 (0)	0 (0)	0 (0)	0 (0)
Never	0 (0)	1 (1.8)	0 (0)	0 (0)	1 (1.8)	0 (0)
Don’t know	0 (0)	0 (0)	1 (1.8)	0 (0)	0 (0)	0 (0)
Observe water containers						
Containers look very clean	50 (89.3)	48 (85.7)	56 (100)	51 (94.4)	56 (100)	52 (96.3)
Containers do not look very clean	6 (10.7)	8 (14.3)	0 (0)	3 (5.6)	0 (0)	2 (3.7)
What do you do with containers you are not currently using?						
Leave them empty as they are	4 (7.1)	3 (5.4)	3 (5.4)	6 (11.1)	2 (3.6)	0 (0)
Turn them upside down	13 (23.2)	13 (23.2)	29 (51.8)	21 (38.9)	33 (58.9)	26 (48.1)
Move them inside	12 (21.4)	23 (41.1)	17 (30.4)	12 (22.2)	15 (26.8)	12 (22.2)
Move them outside	26 (46.4)	18 (32.1)	12 (21.4)	20 (37)	10 (17.9)	16 (29.6)
Don’t have extra containers/Sell	2 (3.6)	1 (1.8)	0 (0)	1 (1.9)	0 (0)	0 (0)
What do you do with waste such as old shells, cans, tires, plastic bottles, and other small containers?						
Bury them	1 (1.8)	0 (0)	0 (0)	0 (0)	0 (0)	0 (0)
Turn them upside down	2 (3.6)	0 (0)	1 (1.8)	2 (3.7)	3 (5.4)	1 (1.9)
Burn them	0 (0)	0 (0)	1 (1.8)	0 (0)	2 (3.6)	1 (1.9)
Move them outside	52 (92.9)	54 (96.4)	55 (98.2)	52 (96.3)	52 (92.9)	52 (96.3)
Sell/Recycle	6 (10.7)	3 (5.4)	0 (0)	0 (0)	1 (1.8)	0 (0)

## Data Availability

The data sets are available and can be accessed on request from the author.

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
