# Peer review of "Effectiveness of Positive Deviance, an Asset-Based Behavior Change Approach, to Improve Knowledge, Attitudes, and Practices Regarding Dengue in Low-Income Communities (Slums) of Islamabad, Pakistan: A Mixed-Method Study"

_insects, 2022, doi:10.3390/insects13010071_

Round 1

Reviewer 1 Report

Authors present a very interesting and convincing study on the use of positive deviance (PD) to foster community engagment in dengue prevention programs.
This represents, in addition, an original approach applied for the first time to dengue control.
The paper is well written and results are clearly presented. Despite this, some key elements seems to be missing in the description of the study design. 
In particular, the full KAP questionnaire should be provided to the reader and, most of all, the PD behaviours identified in the community should be described clearly in
a table or in an additional paragraph. Hence, with with few minor revisions, I strongly agree to publication in your journal.

Minor revision

Abstract
Line 41-42: The PD abbreviation (that should mean "positive deviance I suppose) have to be clearly introducted in the text after its first usage, 
as done at line 83 of the introduction section.

Introduction

Line 52. Aegypti should be written with lower case, as species name. Please, correct.

Line 151. URDU or urdu? Check consistency about this word throught the manuscript.

Materials and methods

Line 152: The full KAP questionnaire, with all the questions about knowledge, attitude and practice, utilized for the analysis should be 
available for the reader as a table or as an additional file.

Line 225: Positive Deviance Inquiry. Which are the main PD behaviours identified by the inquiry? What are the identified PD behaviours that 
have been proposed to the community members of the intervention group?

Results

Line 300: Table 2. Please clarify in the table legend that "n" idicates the number of persons that answer yes to the question. Also clarify 
how attitude and practice were evaluated. In the table there are just scores but any information on the questions.

Author Response

Responses to Reviewer 1:

First of all, thank you so much for your kind review and excellent comments and suggestions which enhanced the quality of the manuscript.  I have tried to incorporate all your comments and suggestion in the document. Please see the detail below:

In particular, the full KAP questionnaire should be provided to the reader and, most of all, the PD behaviours identified in the community should be described clearly in
a table or in an additional paragraph. Hence, with with few minor revisions, I strongly agree to publication in your journal.

KAP survey questionnaire included

PD behavior table in included

Minor revision

Abstract
Line 41-42: The PD abbreviation (that should mean "positive deviance I suppose) have to be clearly introducted in the text after its first usage, 
as done at line 83 of the introduction section.

Done – pl see line 34

Introduction

Line 52. Aegypti should be written with lower case, as species name. Please, correct.

Done – pl see in track changes

Line 151. URDU or urdu? Check consistency about this word throught the manuscript.

Done – urdu throughout the paper

Materials and methods

Line 152: The full KAP questionnaire, with all the questions about knowledge, attitude and practice, utilized for the analysis should be 
available for the reader as a table or as an additional file.

Added a questionnaire

Line 225: Positive Deviance Inquiry. Which are the main PD behaviours identified by the inquiry? What are the identified PD behaviours that 
have been proposed to the community members of the intervention group?

PD behaviour table is added as advised please see line 225

Results

Line 300: Table 2. Please clarify in the table legend that "n" idicates the number of persons that answer yes to the question.

Table 2 legend clarified in track changes,

 English editing was also done as advised

Reviewer 2 Report

The manuscript describes a study of the use of Positive Deviance model to improve knowledge, attitudes and practices for community-based dengue prevention. The study is modest in scope, involving just a one intervention and one control. The inclusion and comparison of some sort of entomological indices would have greatly strengthened the study. In spite of these limitations, however, the results are interesting and suggest a prolonged effect of the intervention on both dengue knowledge and (self-reported) practices. The Positive Deviance model is a potentially important tool for community-based vector control.

The manuscript has several weaknesses, however, that need to be addressed. One is the repeated use of the word ‘slums’. The word is considered insulting, at least in North America, and authors use more neutral expressions like ‘low-income’ or ‘low infrastructure’ communities.  See Slums: The History of a Global Injustice, by Alan Mayne.

Methods: this section needs to be revised to both include some information that is currently missing and also made more concise. There is not sufficient description of the two communities involved in the study, such as spatial area, population and dengue transmission history. Also, given that the population of Islamabad is 95% Muslim, why did the study focus on neighborhoods of the small Christian minority?  There may be a good reason, but the authors need to state it.  In addition, while the authors do explain the statistical calculations that informed the sample size, it is not clear how big the community populations are from which they are making the sampling plan.

The authors provide good information on the two phases of the PD process and intervention, but it’s very long. I wonder if this information could be condensed into a flow chart or some sort of graphic.

Results:  The data presented here seems to be entirely based on the 112 participants’ responses to the same questionnaire three times. Given that, it would be good to include the questionnaire itself in an appendix. Also, more details about exactly what respondents knew (or didn’t know) as well as the reported dengue prevention practices would be useful. The authors refer to separate knowledge, attitude and practice scores, but only explain the criteria for knowledge scores. That should be corrected.

Tables: Table 3, 4 and 5 should be omitted entirely, as they don’t provide much information.

Figures 1, 2 and 3 are great but need error bars.

Discussion: This section is good overall, but the writing is repetitive in some places and could be more concise. The paragraph about the value of small competitions is especially interesting.

Specific Comments: The manuscript is clearly written and easy to understand but has many grammatical mistakes and odd word choices. I began listing them by line, but given the large number, I suggest a revision by a copy editor.

Line 51 – change ‘caused’ to ‘vectored’

Line 52 – the correct capitalization is Aedes aegypti

Line 55 and elsewhere – the phrase ‘dengue disease’ is not typical. Usually it is referred to as dengue or dengue virus.

Line 56 – change to Dengue has expanded dramatically…

Line 93 – change ‘oppose’ to ‘contrast’

There are many other minor changes that should be made to improve the accuracy and precision of the writing.

Author Response

Responses to Reviewer 2:

First of all, thank you so much for your kind review and excellent comments and suggestions which enhanced the quality of the manuscript.  I have tried to incorporate all your comments and suggestion in the document. Please see the detail below:

The manuscript has several weaknesses, however, that need to be addressed. One is the repeated use of the word ‘slums’. The word is considered insulting, at least in North America, and authors use more neutral expressions like ‘low-income’ or ‘low infrastructure’ communities.  See Slums: The History of a Global Injustice, by Alan Mayne.

ANSWER:  I fully understand your point that in most of the cases slum is used as a derogatory word.  However, in Pakistani context, this is quite common and referred to ‘KACHI ABADI’, literally meaning of slum which is always referred to these identified communities.  As you advised, I will mention low-income communities but put the word slum in the bracket (slum).  Most communities lie under low-income in Islamabad, so if we just mentioned low-income communities it will dilute the importance of the study in these special (where minorities live and mostly don’t have the proper housing) communities.  In our context, this is not a derogatory or insulting word/ reference and they themselves refer themselves as people from Kachi-abadies or slums.  

Methods: this section needs to be revised to both include some information that is currently missing and also made more concise. There is not sufficient description of the two communities involved in the study, such as spatial area, population and dengue transmission history. Also, given that the population of Islamabad is 95% Muslim, why did the study focus on neighborhoods of the small Christian minority?  There may be a good reason, but the authors need to state it.  In addition, while the authors do explain the statistical calculations that informed the sample size, it is not clear how big the community populations are from which they are making the sampling plan.

ANSWER:  Great, I have already addressed all these points from lines 125-135 including population, household, justification of selecting the area due to highly affected dengue are in the last dengue outbreak…. And vulnerable community (Christian communities) because of their dwelling which is slums

The authors provide good information on the two phases of the PD process and intervention, but it’s very long. I wonder if this information could be condensed into a flow chart or some sort of graphic.

ANSWER: I have deleted some of the details from 230-233 to make if concise

Results:  The data presented here seems to be entirely based on the 112 participants’ responses to the same questionnaire three times. Given that, it would be good to include the questionnaire itself in an appendix. Also, more details about exactly what respondents knew (or didn’t know) as well as the reported dengue prevention practices would be useful. The authors refer to separate knowledge, attitude and practice scores, but only explain the criteria for knowledge scores. That should be corrected.

ANSWER” yes, as per your advice, I have included the questionnaire and also added three tables no 4, 5, 6  on knowledge, attitude and practices at baseline, mid and endline which provide the viewer better understanding on what behaviours they were engaged in…

Tables: Table 3, 4 and 5 should be omitted entirely, as they don’t provide much information.

ANSWER: Great – I have removed these tables as advised

Discussion: This section is good overall, but the writing is repetitive in some places and could be more concise. The paragraph about the value of small competitions is especially interesting.

ANSWER: I reviewed it again and tried to make it more concise.  I

Specific Comments: The manuscript is clearly written and easy to understand but has many grammatical mistakes and odd word choices. I began listing them by line, but given the large number, I suggest a revision by a copy editor.

Line 51 – change ‘caused’ to ‘vectored’

ANSWER: Changed please see line 55

Line 52 – the correct capitalization is Aedes aegypti

ANSWER: Changed please see line 55

Line 55 and elsewhere – the phrase ‘dengue disease’ is not typical. Usually it is referred to as dengue or dengue virus.

ANSWER: As advised, removed the word ‘disease from 58, 67 and other places in the document

Line 56 – change to Dengue has expanded dramatically…

ANSWER: Changed please see line 59

Line 93 – change ‘oppose’ to ‘contrast’

ANSWER: Changed please see line 96

There are many other minor changes that should be made to improve the accuracy and precision of the writing.

ANSWER: YES – As per your advice, I had it reviewed by a native English speaker who is also an expert in vector control, please see the editing in track changes
